# Palmitic Acid Exerts Anti-Tumorigenic Activities by Modulating Cellular Stress and Lipid Droplet Formation in Endometrial Cancer

**DOI:** 10.3390/biom14050601

**Published:** 2024-05-20

**Authors:** Ziyi Zhao, Jiandong Wang, Weimin Kong, Meredith A. Newton, Wesley C. Burkett, Wenchuan Sun, Lindsey Buckingham, Jillian O’Donnell, Hongyan Suo, Boer Deng, Xiaochang Shen, Xin Zhang, Tianran Hao, Chunxiao Zhou, Victoria L. Bae-Jump

**Affiliations:** 1Department of Gynecologic Oncology, Beijing Obstetrics and Gynecology Hospital, Capital Medical University, Beijing Maternal and Child Health Care Hospital, Beijing 100026, China; zhaoziyi@email.unc.edu (Z.Z.); wangjiandongxy@ccmu.edu.cn (J.W.); kwm1967@ccmu.edu.cn (W.K.); csn2014@mail.ccmu.edu.cn (H.S.); brdeng27@email.unc.edu (B.D.); zhangxin825907@mail.ccmu.edu.cn (X.Z.); 2Division of Gynecologic Oncology, Department of Obstetrics and Gynecology, University of North Carolina at Chapel Hill, Chapel Hill, NC 27599, USA; meredith.newton@unchealth.unc.edu (M.A.N.); wesley.burkett@unchealth.unc.edu (W.C.B.); wenchuan@email.unc.edu (W.S.); lindsey.buckingham@unchealth.unc.edu (L.B.); tianran.hao@wsu.edu (T.H.); 3Lineberger Comprehensive Cancer Center, University of North Carolina at Chapel Hill, Chapel Hill, NC 27599, USA

**Keywords:** palmitic acid, endometrial cancer, lipid droplet, apoptosis, invasion, cellular stress

## Abstract

Epidemiological and clinical evidence have extensively documented the role of obesity in the development of endometrial cancer. However, the effect of fatty acids on cell growth in endometrial cancer has not been widely studied. Here, we reported that palmitic acid significantly inhibited cell proliferation of endometrial cancer cells and primary cultures of endometrial cancer and reduced tumor growth in a transgenic mouse model of endometrial cancer, in parallel with increased cellular stress and apoptosis and decreased cellular adhesion and invasion. Inhibition of cellular stress by N-acetyl-L-cysteine effectively reversed the effects of palmitic acid on cell proliferation, apoptosis, and invasive capacity in endometrial cancer cells. Palmitic acid increased the intracellular formation of lipid droplets in a time- and dose-dependent manner. Depletion of lipid droplets by blocking DGAT1 and DGAT2 effectively increased the ability of palmitic acid to inhibit cell proliferation and induce cleaved caspase 3 activity. Collectively, this study provides new insight into the effect of palmitic acid on cell proliferation and invasion and the formation of lipid droplets that may have potential clinical relevance in the treatment of obesity-driven endometrial cancer.

## 1. Introduction

Endometrial cancer (EC) is the fourth most common cancer in women and the most common gynecologic malignancy in the United States, with 66,200 new cases and 13,030 disease-related deaths projected in 2023 [1]. Although usually detected at earlier and more favorable stages with a better prognosis, EC diagnosed at an advanced stage is prone to recurrence and lacks optimal adjuvant therapies and strategies [2]. Obesity is a well-established risk factor for the development of EC and may pose a threat to treatment efficacy and prognosis [3,4]. Increased adipose tissue, particularly visceral adipose tissue, is an endocrine organ that regulates metabolism and inflammatory responses and produces bioactive substances associated with cell growth, differentiation, and angiogenesis, all of which have been implicated in EC carcinogenesis [5]. Recent studies have found that ECs display specific changes in different aspects of lipid metabolism, including fatty acid metabolism, under obese and lean conditions, which may be clinically relevant to EC progression and response to chemotherapy [6,7,8,9].

Cancer cells create a metabolic environment that is conducive to cancer cell growth by altering the metabolism of lipids, carbohydrates, and proteins, allowing cancer cells to effectively maintain the functionality of structures and functions in the environment [10]. Fatty acids (FAs), as chemically heterogeneous compounds, are intimately involved in the synthesis and metabolism of lipids, are required for energy storage, membrane structure, and signaling precursor molecules, and exhibit diverse functions in carcinogenesis, progression, and chemotherapy in cancer [11]. Recent studies have found that the production of monounsaturated fatty acids (MUFAs) from saturated fatty acids (SFAs) is an important indicator of regulating the fluidity, functionality, and flexibility of biological membranes, and deregulation of the ratio of MUFAs to SFAs is essential for determining cell proliferation, invasiveness, and signaling cascades that control the proliferation of tumor cells [12,13,14]. The FA composition of membrane phospholipids is continuously remodeled by the availability of free fatty acids, the enzymatic activity of phospholipases, stress conditions, and metabolic diseases, suggesting that changes in the environmental conditions of tumor growth in vivo or alterations in the proportion of ingested FAs can affect the proliferative activity of cancer cells [15,16,17]. The high rate of fatty acid synthesis in cancer cells is initiated by the enzymatic system of fatty acid synthase (FAS) to form abundant 16-carbon palmitic acid (PA), the most widely recognized saturated long-chain fatty acid found in the human body, which represents 65% of SFA in the body and 28–32% of total serum FA as well as 10–20% of human dietary fat intake [18,19]. Biochemical pathways for MUFA synthesis rely primarily on the desaturation or elongation of PA by the action of so-called elongases and desaturases in the body in addition to dietary supplementation. Several MUFAs such as sapienic and palmitoleic acids significantly influence membrane composition and properties and change the expression of growth factors and signaling cascades in breast cancer cells [13,20]. Accumulating evidence indicates that desaturases, including stearoyl-CoA desaturases-1 (SCD1), have emerged as key players in the regulation of metabolic and signaling pathways that support the biochemical and biological phenotype of cancer cells, and targeting SCD1 has the potential to develop new cancer therapeutics [21,22]. In addition to the palmitoylation of PA involved in the functional regulation of multiple genes including oncogenes and tumor suppressor genes, it is generally believed that PA acts as an energy source to promote cell growth through β-oxidation and participates in the synthesis of cancer cell membranes by functioning as a signaling molecule in cancer cells [23,24]. Increased dietary PA intake or elevated circulating levels of plasma phospholipid PA have been found to be strongly associated with the risk of breast cancer [25,26]. However, there is growing evidence that PA inhibits cancer cell proliferation, induces cellular stress and apoptosis, causes cell cycle arrest, and impairs cell invasiveness, either directly or indirectly in various types of preclinical models [18,25,27]. Importantly, PA potently inhibited cell viability and induced apoptosis along with a concomitant increase in PA composition in neuroblastoma cell membranes, without affecting the overall polyunsaturated content. The combination of PA with oleic and arachidonic acids resulted in changes in cell membrane lipidome with suppression of caspase activation and maintenance of cell viability, demonstrating an important role for membrane FA reorganization in suppressing cell growth and the potential for dietary FAs, including PA, to intervene in cancer [28,29].

Given that PA is the most common SFA in the human diet and in the human body, and that obesity and metabolic disorders are the most important risk factors for the carcinogenesis and progression of EC, it is necessary to evaluate the function of PA in EC cell proliferation and tumor growth [4,18]. Understanding the role of PA in cell and tumor growth in EC is critical for the design of novel treatment and prevention strategies using this dietary supplementation. In this study, we aimed to characterize the effects of PA on cell proliferation and tumor growth in EC cell lines and a transgenic mouse model of EC. Our results show that PA inhibits EC cell proliferation and invasion, increases the formation of cellular lipid droplets, and reduces tumor growth.

## 2. Materials and Methods

### 2.1. Cell Culture and Reagents

Two human endometrial cancer cell lines, Ishikawa (RRID: CVCL_2529) and ECC-1 (RRID: CVCL_7260), were used in this study. Both cell lines were obtained and authenticated from the Cell bank in Lineberger Cancer Center, the University of North Carolina at Chapel Hill. During the experiments, we regularly detected mycoplasma contamination in cell cultures every six months by Mycoplasma Detection Assays (InvivoGen, San Diego, CA, USA). Ishikawa cells were maintained in DMEM/F12 medium with 10% fetal bovine serum (FBS), and ECC-1 cells were maintained in RPMI 1640 medium with 10% FBS. All media were supplemented with penicillin (100 U/mL), streptomycin (100 ug/mL), and L-glutamine (2 mM/mL). The cells were cultured in a humidified 5% CO_2_ incubator at 37 °C. PA was obtained from Sigma (St. Louis, MO, USA). T863 and PF-06424439 were from Cayman (Ann Arbor, MI, USA). Antibodies for Western blotting including PERK (#5683), IRE1-α (#2394), ATF4 (#11815), Bip (#3177), CLPP (#14181), Calnexin (#2679), BCL-XL (#2764), MCL-1 (#5453), PARP (#9532), α-tubulin (#2144), β-actin (#3700), Snail (#3879), VEGF-C (#2445), Vimentin (#5741), N-cadherin (#13116), MMP-9 (#13667), p-s6 (#4858), s6 (#2217), p-Akt (#4060), Akt (#4691), p-42/44(#4370), 42/44 (#4695), p38 (#4511), p-4E-BP1 (#2855), p-ACC (#11818), ATGL(#2439), FASN (#3180), ACSL-1 (#9189), Lipin-1 (#14906), CPT1A (#12252), HK I (#2024), and HK II (#2867) were obtained from Cell Signaling Technology (Beverly, MA, USA). Glut1(#A6982), Glut4 (#A7637), LDHA (#A1146), DGAT1 (#A6857), and DGAT2 (#13891) were purchased from Abclonal Technology (Woburn, MA, USA). Antibodies for Immunochemistry including Ki-67(#12202), BCL-XL (#2764), p-ACC (#11818), ATGL (#2439), p-44/42 (#4370), p-s6 (#4858), and VEGF-R2 (#9698) were obtained from Cell Signaling Technology. PERK (#SC377400) was purchased from Santa Cruz Technology. Details of antibody use are provided in Appendix A. Enhanced chemiluminescence Western blotting detection reagents were obtained from Amersham (Arlington Heights, IL, USA). All other chemicals were purchased from Thermo Fisher Scientific (Waltham, MA, USA).

### 2.2. Preparation of BSA-Bound PA

PA was prepared as described previously [30]. Briefly, 0.053 g of PA (Sigma-Aldrich, P0500, St. Louis, MO, USA) powder was dissolved in 210 μL NaOH solution (0.2 M) by heating at 75 °C in a shaking water bath for 15 min until the PA-NaOH solution turned clear. Then, 840 μL of 30% fatty-acid-free BSA (Sigma-Aldrich, A9576, St. Louis, MO, USA) was added, and the combination was vortexed for 10 s followed by a further 10-minute incubation at 55 °C. The final concentration of PA in the BSA-PA complex was 20 mM at a ratio of PA:BSA = 5:1. The BSA-bound PA was filtered through a 0.22 μm filter and stored as aliquots in a −20 °C freezer. When the cells were treated with PA, the control cells were treated with an equal volume of BSA solution.

### 2.3. Primary Culture of Human-Derived EC

The collection of human EC tissue was approved by the Institutional Review Board of the University of North Carolina at Chapel Hill and performed at UNC Hospital (IRB #10-1206). After informed consent was obtained from all individual patients, seventeen tumor tissues were collected from patients with EC at the time of hysterectomy (Appendix A). The pathological diagnosis of EC was made by pathologists at UNC-CH. Cube samples of 5 × 5 mm were placed in a DMEM/F12 culture medium containing antibiotics and transferred to the laboratory for primary cultures. The tissues were washed with PBS three times and then digested in 0.5% collagenase IA, 0.1% DNase, and 100 U/mL penicillin and streptomycin for 30–60 min in a 37 °C water bath with shaking. After washing twice with PBS, the tumor cells (2 × 10^4^/per well) were seeded in 96-well plates and cultured in DMEM/F12 containing 10% FBS. Cell proliferation was measured by an MTT assay 72 h after treatment with PA.

### 2.4. Lkb1^fl/fl^p53^fl/fl^ Transgenic Mouse Model of EC

In order to investigate the effect of PA on tumor growth in vivo, we utilized the *Lkb1^fl/fl^p53^fl/fl^* mouse model of EC [31]. This animal study was approved by the Institutional Animal Care and Use Committee (IACUC) of the University of North Carolina at Chapel Hill (protocol # 21-209). The mice were housed at our animal facility with a 12 h light, 12 h dark cycle and allowed free access to food (50 IF/6F diet, PecoLab, Saint Paul, MN, USA) and water. Six- to eight-week-old female mice were injected with 5 µL of 2.5 × 10^11^ P.F.U of recombinant adenovirus Ad5-CMV-Cre (Transfer Vector Core, University of Iowa, Iowa City, IA, USA) in the left uterine horn. The mice were randomly divided into two groups as follows: a control group and a PA treatment group (18 mice per group). After 8 weeks following injection with Ad5-CMV-Cre, the mice were treated with PA (10 mg/kg, 100 µL per mouse for oral gavage, daily) or vehicle (100 μL, 2.5:1 ratio of water and 0.2 N-NaOH, daily) for 4 weeks. This dose of PA has been confirmed to effectively inhibit cancer growth, including liver and pancreatic cancers, without causing any side effects in animal studies [32,33,34]. Mice were weighed twice a week throughout treatment. During PA treatment, the mice exhibited normal activity and behavior, and no mice died in either group. All mice were sacrificed by CO_2_ asphyxiation after 4 weeks of treatment. EC tumors, visceral fat tissues, liver, and blood were carefully harvested and recorded.

### 2.5. Cell Viability Assay

Cell viability was assessed using an MTT assay. In brief, the Ishikawa and ECC-1 cells were plated at a concentration of 6.0 × 10^3^ cells/well and 4.0 × 10^3^ cells/well, respectively, and cultured for 24 h. The cells were then treated with various concentrations of PA for 72 h. MTT (5 mg/mL) was added into each well for 1 h. Subsequently, the culture medium was discarded, and 100 µL dimethyl sulfoxide (DMSO) was added to each well to terminate the reaction. The optical density (OD) value of each sample was detected at a wavelength of 562 nm with a microplate reader (Tecan, Morrisville, NC, USA). The effect of PA on cell proliferation was calculated as a percentage of control. The AAT Bioquest calculator was used to determine IC50 values. Each experiment was performed at least three times to assess the consistency of the results.

### 2.6. Colony Assays

The Ishikawa and ECC-1 cells were harvested from a stock culture and plated at a density of 400 cells/well in six-well plates. After the attachment of cells to the wells (approximately 2 to 3 h after incubation), the cells were treated with various concentrations of PA. Thirty-six hours after PA treatment, the media were replaced with fresh media, and the medium was changed every 3 days. After culturing for 10–12 days, each well was washed with PBS and stained with a mixture of 6.0% glutaraldehyde and 0.5% crystal violet. The clones were imaged and quantified using Image J software (V1.8.0, National Institutes of Health, Bethesda, MD, USA).

### 2.7. Preparation of Cell-Based Functional Assays

In addition to the wound healing and transwell assays in the following cell-based functional assays, the Ishikawa and ECC-1 cells were cultured in their regular media and treated with 1, 100, and 250 μM PA for different times. Control cells were treated with an equal volume of BSA solution. All experiments were performed at least three times to assess the consistency of the response.

### 2.8. Glucose Uptake Assay

Glucose uptake was determined by the 2-NBDG Glucose Uptake Assay (BioVision, Milpitas, CA, USA). Briefly, the Ishikawa and ECC-1 cells on 96-well plates were treated with PA for 16–18 h. The cells were washed with HBSS and incubated in a glucose-free medium with 2-[N-(7-nitrobenz-2-oxa-1,3-diazol-4-yl) amino]-2-deoxy-D-glucose (2-NBDG, 100 μg/mL) for 15 min. After washing the plates with Hanks’ Balanced Salt Solution (HBSS) twice, the fluorescence intensity (Ex/Em = 465/540) of cellular 2-NBDG in each well was measured using a Tecan plate reader. The experiments were performed in triplicate and repeated three times.

### 2.9. Lactate Assay

The Ishikawa and ECC-1 cells were cultured in six-well plates at 2.5 × 10^5^ cells/well overnight and then treated with PA for 24 h. The media were collected to assess lactate production using the BioVision lactate assay kit (Milpitas, CA, USA) and following the manufacturer’s instructions. The OD values were determined at 450 nm with a Tecan plate reader. In the meantime, the cells in each well were collected by Trypsin-EDTA (0.25%), and the cell density was counted using a Cellometer (Nexcelom Bioscience, Waltham, MA, USA). The lactate levels in each well were normalized to the cell number.

### 2.10. ATP Assay

The luminometric ATP assay kit (AAT bioquest, Sunnyvale, CA, USA) was used to investigate the effects of PA on the cellular production of ATP in EC cells. The Ishikawa and ECC-1 cells were seeded at 5 × 10^3^/well in 96-well plates overnight and then treated with PA for 24 h. Then, 100 µL ATP detecting solution was added into each well, mixed, and incubated for 20 min at room temperature. The luminescence intensity was immediately measured on a Tecan plate reader. After detecting ATP, 5 μL MTT solution (5 mg/mL) was added to each well, and the plate was incubated at 37 °C for 1 h. The ATP levels were normalized according to the viable cell counts measured by the MTT assays.

### 2.11. Cleaved Caspase 3, 8, and 9 ELISA Assays

The Ishikawa and ECC-1 cells were plated in 6-well plates at the concentration of 2.5 × 10^5^ cells/well and 3.5 × 10^5^ cells/well, respectively. After 24 h, the cells were exposed to PA for 14 h and 150–180 µL of 1× caspase lysis buffer was added to each well. Cell lysates were collected, and protein concentrations were determined via a BCA assay (Thermo Fisher). A total of 30 ug lysate was added to each well of a black 96-well plate. Reaction buffer with caspase 3, 8, or 9 substrates was added to each well and mixed with lysate at 37 °C for 20 min. The fluorescence intensity for cleaved caspase 3 (Ex/Em = 400/505), cleaved caspase 8 (Ex/Em = 376/482), cleaved caspase 9 (Ex/Em = 341/441) were recorded using a Tecan microplate reader.

### 2.12. Reactive Oxygen Species (ROS) Assay

The Fluorometric Intracellular Total ROS Activity Assay Kit (AAT Bioquest, Sunnyvale, CA, USA) was used to detect alterations in the production of ROS. The Ishikawa and ECC-1 cells (6.0 × 10^3^ cells/well) were seeded into black 96-well plates. After 24 h, the cells were treated with PA and allowed to incubate for 12 h at 37 °C to induce ROS generation. DCFH-DA (15 µM) was then applied to the cells and allowed to incubate for 30 min. The fluorescence intensity was measured at Ex/Em 485/530 nm using a Tecan microplate reader.

### 2.13. Mitochondrial Membrane Potential Assays

Mitochondrial membrane potential was analyzed with specific fluorescent probes for JC-1 and TMRE (AAT Bioquest, Sunnyvale, CA, USA). The Ishikawa and ECC-1 cells were plated in 96-well plates overnight and treated with PA for 14 h. The cells were then treated with 2 µM JC-1 or 800 µM TMRE for 30 min at 37 °C. The microplate reader measured fluorescence intensity at Ex/ Em = 480/590 nm for the JC-1 assay and Ex/Em = 549/575 nm for the TMRE assay, respectively.

### 2.14. Adhesion Assay

Each well in a 96-well plate was coated with 100 µL laminin-1 (10 μg/mL) and incubated at 37 °C for 1 h. The fluid was then aspirated, and 200 μL of blocking buffer was added to each well for 45 to 60 min at 37 °C and then washed with PBS while chilling on ice. To each well, 1.2 × 10^4^ cells were added with PA. The plate was then allowed to incubate at 37 °C for 1.5–2 h. After this period, the medium was aspirated, and the cells were fixed by adding 100 µL of 5% glutaraldehyde and incubating for 30 min at room temperature. Adhered cells were then washed with PBS and stained with 100 µL of 0.1% crystal violet for 30 min. The cells were then washed repeatedly with water, and 100 µL of 10% acetic acid was added to each well to solubilize the dye. The plate was shaken for 5 min, and absorbance was measured at 562 nm using a Tecan microplate reader.

### 2.15. Transwell Assay

Cell invasiveness was quantified by a modified Boyden chamber assay (Corning Inc., Corning, NY, USA). The Ishikawa and ECC-1 cells were starved in 1% FBS for 12–14 h, seeded on top of a Matrigel invasion chamber (8 μm pore size) with a density of 3 × 10^4^, and then 1, 100, and 250 μM PA was added to the top chambers. The lower chambers were filled with regular medium. The plates were cultured for 4–6 h to allow cell invasion into the lower chambers. After removing the top chambers and washing the lower chambers with PBS, 100 µL of calcein AM solution (Invitrogen, Carlsbad, CA, USA) was applied to the lower chambers and incubated for 30–60 min at 37 °C. The fluorescence intensity of the lower chambers was measured by a Tecan reader at Ex/Em 485/520 nm.

### 2.16. Wound Healing Assay

The Ishikawa and ECC-1 cells were plated at 4.0 × 10^5^ cells/well in six-well plates and cultured overnight. A uniform wound was created by a 20 μL pipette. The cells were washed and then treated with PA for 48 h in regular media with 1% FBS. Photos were taken at 24 and 48 h after scratching, and the width of the wound was measured and analyzed with ImageJ software (V1.8.0, National Institutes of Health, Bethesda, MD, USA). The percent closure of the scratch was measured by comparison to control cells.

### 2.17. Fatty Acid Oxidation (FAO) Assay

FAO activity was detected by an FAO enzyme assay (Biomedical Research Service, State University of New York at Buffalo). The Ishikawa and ECC-1 cells were plated in 6-well plates at a density of 2.5 × 10^5^ cells/well, cultured overnight at 37 °C, and then treated with PA for 24 h. The cells were harvested with ice-cold 1× Sample Buffer and the supernatant was collected after centrifugation. Protein concentrations were measured by a BCA kit. After equalizing the concentrations of protein, 20 µL of each sample was treated with 50 µL of FAO Assay Solution or 50 µL of control solution. The mixed solutions were added to a 96-well plate. The plate was kept in a non-CO_2_ incubator at 37 °C for 60 min. The OD value was measured by a Tecan reader at a wavelength of 492 nm. A blank reading was subtracted from the sample reading.

### 2.18. Oil Red O Staining Assay

The Ishikawa and ECC-1 cells were planted in 12-well plates at a density of 1.2 × 10^5^ cells/well and grown for 24 h. Both cell lines were treated with PA for 24 h, and then the cells were stabilized in fresh formalin for 1 h. Oil Red O (Sigma Aldrich, St. Louis, MO, USA) and Harris Modified Hematoxylin were added to the wells for staining. Photos were taken by a Thermo Scientific Invitrogen EVOS Microscope (Thermo Fisher Scientific, Waltham, MA, USA).

### 2.19. Western Immunoblotting

Following the treatment of Ishikawa and ECC-1 cells with different concentrations of PA for 6 to 36 h, total proteins were extracted from both cell lines using RIPA buffer containing protease and phosphatase inhibitors (Thermo Fisher). A BCA protein assay kit (Thermo Fisher) was used to determine the protein concentration. Equal amounts of protein were separated by gel electrophoresis and transferred onto a PVDF membrane. The membranes were blocked with 5% non-fat dry milk and then incubated with 1:1000 to 3000 dilutions of primary antibody overnight at 4 °C. The appropriate secondary antibody was incubated with the membrane for 1 h at room temperature after washing with TBS-T. Immunoblots were developed using an enhanced chemiluminescence detection buffer, and bands were visualized with the Bio-Rad Imaging System (version 3.0.1, Hercules, CA, USA). After development, the membranes were stripped or washed and re-probed using α-tubulin or β-actin antibodies. Each experiment was repeated at least twice to assess the consistency of the results.

### 2.20. Immunohistochemistry (IHC)

Endometrial tumor slides (4 µm) from the *Lkb1^fl/fl^p53^fl/fl^* mice were used for the IHC study. Following deparaffinization and rehydration, endogenous peroxidase was blocked in 3% H_2_O_2_ for 10 min. Antigen retrieval was performed in a microwave oven using a citrate buffer (sodium citrate, pH 6.0). The slides were incubated with protein block solution (Dako) for 1 h, and then the primary antibodies were added overnight in a cold room (six slides per group). The slides were then washed and incubated with appropriate secondary antibodies at room temperature for 1 h. After removing the secondary antibody, the specific staining was visualized using the Signal Stain Boost Immunohistochemical Detection Reagent (Cell Signaling Technology, Danvers, MA, USA), according to the manufacturer’s instructions. Individual slides were scanned using Motic (Feasterville, PA, USA), and digital images were analyzed for target protein expression using ImagePro software (V2, Vista, CA, USA). The IHC scoring formula was used to express the expression level of the target protein.

### 2.21. HE Staining

The livers of EC mice were carefully harvested after sacrifice and embedded in paraffin. The liver slides (4 μm) were stained with Hematoxylin to visualize the cell nucleus and Eosin to visualize the cytoplasm. Individual slides were scanned using Motic (Feasterville, PA, USA), and digital images were analyzed using ImagePro software (Vista, CA, USA).

### 2.22. Serum Vascular Endothelial Growth Factor (VEGF) and Triglyceride (TG) Assays

Serum VEGF and TG levels after treatment with PA were detected using an R&D VEGF ELISA Kit (Catalog #: MMV00, Minneapolis, MN, USA) and a BioVision Triglyceride Quantification Kit (Catalog #: ab65336), according to the manufacturers’ directions. Each sample from the PA treatment and control groups was measured in triplicate. The plates were read at 570 nm for VEGF and TG measurements using a Tecan plate reader.

### 2.23. Statistical Analysis

All data are reported as mean ± SD from three independent assays. The Student’s *t*-test and One-way ANOVA test were used in this study. GraphPad Prism 8 (La Jolla, CA, USA) statistical software was employed to calculate the comparisons. All tests were two-sided with *p* < 0.05 considered significant.

## 3. Results

### 3.1. PA Inhibited Cell Proliferation and the AKT/mTOR, MAPK, and p38 Pathways in EC Cells

To investigate whether PA affects EC cell proliferation, the Ishikawa and ECC-1 cells were subjected to an MTT assay. As shown in Figure 1A, both cell lines treated with different concentrations of PA showed a significant decrease in cell viability, and the IC50 values for the Ishikawa and ECC-1 cells were 348.2 ± 30.29 µM and 187.3 ± 19.02 µM, respectively, after 72 h of treatment. When both cell lines were treated with a physiologic concentration of 250 µM PA for 72 h [18], cell proliferation was decreased by 25.7% in the Ishikawa cells and 65.1% in the ECC-1 cells relative to the BSA control. Consistent with the MTT assays, the colony assays showed that after treatment with 250 µM PA for 36 h and continued cell culture for 10–12 days, the colony formation of the Ishikawa and ECC-1 cells was reduced by approximately 40.6% and 70.8%, respectively, compared with their respective control cells (*p* = 0.025) (Figure 1B). Since primary cell cultures from patient-derived tumors may be better predictors of anti-tumorigenic activity of cytotoxic agents than cancer cell lines, seventeen total primary cell cultures of EC were cultured with varying concentrations of PA for 72 h. The MTT assay revealed that the primary culture cells exhibited variable responses to PA treatment. Nine of the seventeen primary cell cultures displayed at least 10% inhibition of cell proliferation at 250 µM PA (Figure 1C and Appendix A).

To investigate the role of the AKT/mTOR, MAPK, and p38 pathways in the PA-mediated inhibition of cell proliferation, the Ishikawa and ECC-1 cells were treated with different concentrations of PA for 24 h, and a Western blotting assay was used to detect changes in these pathways using antibodies against the specific phosphorylated forms of AKT, MAPK, and p38. The results showed that the expression of phosphorylated AKT, S6, 4E-BP1, p38 and p42/44 was downregulated in both cell lines following 250 μM PA treatment (Figure 1D). These results suggest that PA reduced cell viability via effects on the AKT/mTOR, MAPK, and p38 pathways.

### 3.2. PA Induced Apoptosis in EC Cells

To assess the effects of PA on inducing apoptotic activity in EC cells, the Ishikawa and ECC-1 cell lines were treated with different concentrations of PA for 14 h, and the apoptotic activity of PA-treated cells was measured by cleaved caspase 3, caspase 8, and caspase 9 ELISA assays. As shown in Figure 2A–C, 100 µM and 250 µM treatments significantly increased cleaved caspase 3, 8, and 9 activities in both cell lines. Treatment of EC cells with 250 µM PA increased cleaved caspase 3 by 1.92 times in the Ishikawa cells and 2.35 times in the ECC-1 cells (*p* = 0.028). The Western blotting results showed that PA reduced the expression of Bcl-xL and Mcl-1 and increased cleaved PARP protein expression in a dose-dependent manner in both cell lines after treatment with PA for 14 h (Figure 2D). To evaluate the role of mitochondrial apoptotic pathways in PA-induced apoptosis, both cell lines were pre-treated with Z-VAD-FMK (10 µM, a pan-caspase inhibitor) for 1.5 h and then treated with 250 µM PA for 14 h for the caspase 3 assay and 72 h for the MTT assay. Pre-treatment with Z-VAD-FMK blocked PA-induced cleaved caspase 3 activity and significantly rescued PA-induced apoptosis relative to the control-treated cells (*p* = 0.041) (Figure 2E,F), indicating that that PA-induced apoptosis depends on the extrinsic and intrinsic mitochondrial apoptotic pathways in EC cells.

### 3.3. PA Induced Cellular Stress in EC Cells

To investigate the role of cellular stress on PA-induced apoptosis in EC cells, cellular ROS levels were assessed using a DCFH-DA assay. Treatment of both cell lines with 100 µM or 250 µM PA significantly increased ROS production (Figure 3A). PA increased ROS production by 1.43 times in the Ishikawa cells and 1.53 times in the ECC-1 cells at a dose of 250 µM compared with the control groups (*p* = 0.026). To further confirm whether the PA-induced increase in ROS is related to mitochondrial function, the JC-1 and TMRE ELISA assays were employed to detect changes in mitochondrial membrane potential (ΔΨm). In the Ishikawa cells, 100 µM PA appeared to decrease ΔΨm by TMRE but not the JC-1 assay; however, 250 µM PA statistically decreased ΔΨm by both the JC-1 and TMRE assays in both cell lines (*p* = 0.034). In the ECC-1 cells, both 100 µM and 250 µM PA treatments induced a loss in ΔΨm in both assays after 14 h of treatment relative to control cells (*p* = 0.015) (Figure 3B,C). The effect of PA on the expression of ER stress-related proteins was detected by Western blotting analysis. The results indicated that treatment of cells with 100 or 250 µM of PA for 24 h up-regulated the protein expression of PERK, ATF4, BiP, IRE1-α, and CLPP in both cell lines (Figure 3D). These results indicate that cellular stress is also responsible, in part, for the anti-proliferative effect of PA in EC cell lines.

### 3.4. PA Reduced Adhesion, Migration, and Invasion in EC Cells

To investigate the influence of PA on adhesion and invasion in EC cells, the laminin-1 adhesion assay, the transwell assay, and the wound healing assay were performed in the Ishikawa and ECC-1 cells. In the Ishikawa cells, PA reduced cell adhesion and invasion at an optimal concentration of 250 µM, while in the ECC-1 cells, in addition to inhibiting cell adhesion and invasion at a dose of 250 µM PA, 100 µM PA also significantly decreased cell adhesive and invasive capacity (*p* = 0.018) (Figure 4A,B). Similarly, the results of the wound healing assay showed that 100 µM or 250 µM PA suppressed the migration of the Ishikawa and ECC-1 cells. After treating the cells with 100 and 250 µM PA for 48 h, the migration of Ishikawa cells was inhibited by 12.4% and 44.4%, respectively, and the migratory potential of ECC-1 cells was inhibited by 31.1% and 51.9%, respectively, relative to the control cells (*p* = 0.026) (Figure 4C). To determine whether the epithelial–mesenchymal transition (EMT) process involved PA-mediated invasion and migration, the Ishikawa and ECC-1 cells were incubated overnight with different concentrations of PA, and Western blot assays were used to determine the expression of EMT and angiogenic markers. The results demonstrated that high concentrations of PA (250 µM) reduced the expression of VEGF-C, MMP9, Snail, Vimentin, and N-Cadherin in both cell lines (Figure 4D). Overall, our results suggest that PA has the potential to inhibit adhesion and invasion in EC cells.

### 3.5. Cellular Stress Modulated PA-Induced Migration and Invasion

To verify the role of cellular stress in the migration and invasion induced by PA, the Ishikawa and ECC-1 cells were pre-incubated with 0.1 mM NAC (N-acetyl-l-cysteine) for 3 h and then treated with 250 and 500 µM PA at the indicated times. Pre-treatment of both cell lines with NAC effectively abolished PA-associated ROS levels. Moreover, NAC partially reversed the inhibitory effect of PA on cell proliferation and attenuated PA-induced apoptosis in both cell lines (*p* = 0.031) (Figure 5A,B). In addition, NAC was able to alleviate the effects of PA on the inhibition of wound healing and decreased invasive capacity in both cell lines (*p* = 0.029) (Figure 5C,D). The Western blotting results demonstrated that pre-treatment with NAC decreased PA-mediated downregulation of Bcl-xL, Mcl-1, Calnexin, and PERK in both cell lines (Figure 5E). These results suggest that cellular stress pathways are a potential trigger that controls the effects of PA on cell proliferation, apoptosis, and invasion in EC.

### 3.6. PA Caused Lipid Accumulation and Increased Lipogenesis in EC Cells

Given that exogenous fatty acids increase lipid droplet (LD) formations and LDs are thought to be involved in multiple metabolic processes in cancer cells, we investigated the regulatory role of PA in the formation of LDs by Oil Red O staining in the EC cell lines. PA increased LD accumulation in a dose-dependent manner in both the Ishikawa and ECC-1 cells after 24 h of treatment (*p* = 0.033) (Figure 6A). Furthermore, treatment of both cell lines with 100 µM PA increased Oil Red O staining in a time-dependent manner compared with untreated cells, reaching a maximum at 9 h for the Ishikawa cells and at 18 h for the ECC-1 cells (*p* = 0.023) (Figure 6B). Phosphorylation of ACC and adipose triglyceride lipase (ATGL) was observed by a Western blotting assay after treatment in both cell lines with 100 μM PA, in a time-course fashion. PA significantly increased ACC phosphorylation and ATCL expression in both cell lines within 24 h of 100 µM PA treatment (Figure 6C). Since LDs are the main intracellular organelles that store triglycerides (TGs) and other neutral lipids, intracellular TG levels were determined by an ELISA assay. Treatment of cells with 100 µM and 250 µM PA overnight also increased intracellular TG production by 20.9% and 54.1% in the Ishikawa cells and 46.6% and 66.4% in the ECC-1 cells, respectively (*p* < 0.05), accompanied by increased the expression of ATGL, long-chain acyl-CoA synthetase-1 (ACSL-1), and fatty acid synthase (FASN) in both cell lines (Figure 6D,E). To evaluate the influence of PA on mitochondrial fatty acid oxidation (FAO) in EC cells, both cell lines were treated with 100 µM PA in a time-course manner. Interestingly, increased expression of CTP1A and activity of FAO were observed in both cell lines within 18 h of treatment, while CPT1A expression and FAO activity were decreased when the cells were treated with 100 and 250 µM PA for 24 h (Figure 6F,G). Taken together, these results demonstrate that PA is effective in increasing LD formation and de novo fatty acid synthesis in EC cells, supporting the concept of simultaneous fatty acid synthesis and FAO in PA-treated EC cells [35,36].

Since fatty acids can stimulate cancer cell metabolic reprogramming, we next examined the effects of PA on glycolysis metabolism in EC cells. PA treatment at doses of 100 and 250 µM for 24 h significantly reduced glucose uptake and lactate production in both cell lines (*p* = 0.043) (Figure 6H). In further support of this, high concentrations of PA reduced the expression of Glut1, Glut4, LDHA, Hexokinase (HK) I, and Hexokinase II in the Ishikawa and ECC-1 cells, as assessed by Western blotting analysis (Figure 6I). Finally, 250 µM PA reduced cellular ATP production by 15.3% in the Ishikawa cells and 19.3% in the ECC-1 cells, respectively, compared with the control cells after 24 h of treatment (*p* = 0.014) (Figure 6J). These data support the existence of feedback regulation between PA metabolism and glycolytic activity.

### 3.7. Inhibition of LD Formation Increased the Sensitivity to PA in EC Cells

To investigate the role of LD formation in PA-mediated growth inhibition, we selected two small inhibitors, T863 and PF-06424439, to block the functions of DGAT1 and DGAT2, which catalyze the final committed step in mammalian TAG biosynthesis [37,38]. To reduce the effect of PA-induced inhibition of cell proliferation on LD formation, the Ishikawa and ECC-1 cells were treated with 150 µM PA with or without 5 µM T863 or 5 µM PF-06424439 and their combinations for 24 h. As shown in Figure 7A,B, targeting either DGAT1 by T863 or DGAT2 by PF-06424439 reduced LD formation, and T863 was shown to be more effective than PF-06424439 in reducing LD formation in both EC cells (*p* = 0.019). Importantly, PA in combination with T863 and PF-06424439 completely blocked LD formation. The MTT analysis showed that blocking DGAT1 by T863 but not DGAT2 by PF-06424439 effectively increased PA sensitivity in the inhibition of cell proliferation, but the combination of T863, PF-06424439, and PA produced the greatest cytostatic effects in both cell lines (*p* = 0.036) (Figure 7C). Consistent with our MTT results, the depletion of LDs by combination treatment with T863 and PF-06424439 significantly increased cleaved caspase 3 activity in PA-treated EC cells (*p* = 0.021) (Figure 7D). These results suggest that LDs have a cytoprotective effect on EC cells in the presence of high concentrations of PA.

### 3.8. PA Inhibited Tumor Growth and Increased Lipogenesis in the Lkb1^fl/fl^p53^fl/fl^ Mouse Mode of EC

To determine the anti-cancer activity of PA on tumor growth, *Lkb1^fl/fl^p53^fl/fl^* mice (18 mice/per group) were treated with 10 mg/kg PA or vehicle daily through oral gavage for 4 weeks. The body weight of each mouse was measured twice a week. During the treatment period, the mice exhibited normal activity and no significant changes in body weight (Figure 8A). There were no changes in serum triglycerides (TGs) and liver weights in the mice after completion of treatment compared with the controls (Figure 8B,C). However, H&E staining of liver sections showed a significant increase in ballooned hepatocytes in PA-treated mice, indicating that PA increased the accumulation of fat in hepatocytes (Figure 8D). PA treatment significantly reduced tumor weight by 43.2% compared with the control mice (*p* = 0.022) (Figure 8E). Finally, IHC staining was applied to analyze the difference in cell proliferation between groups after PA treatment. PA inhibited the expression of Ki-67 by 52.4% compared with the control group (*p* = 0.025) (Figure 8F). These results suggest that PA effectively inhibited EC cell proliferation and tumor growth in vitro and in vivo.

To evaluate the mechanisms by which PA inhibits tumor growth in vivo, the expression of Bcl-xL, PERK, VEGF-R2, phospho-ACC, ATGL, phospho-42/44, and phos-S6 was examined by IHC, and serum VEGF production was determined by an ELISA assay in the endometrial tumors of *Lkb1^fl/fl^p53^fl/fl^* mice. PA did not affect the levels of serum VEGF after 4 weeks of treatment (Appendix A); however, the expression of Bcl-xL, VEGF-R2, phospho-42/44, and phospho-S6 in the endometrial tumors of PA-treated mice was notably decreased, while the expression of PERK was increased when compared with control mice (*p* = 0.015) (Figure 8G). To observe the effect of PA on lipid metabolism in vivo, a Western blotting assay was used to determine the changes in fatty acid synthesis in EC tissues from *Lkb1^fl/fl^p53^fl/fl^* mice. The results showed that PA treatment increased the expression of FASN, Lipin-1, and phosphorylated-ACC and decreased the expression of phosphorylation of S6 (Figure 8H). IHC also confirmed similar results, indicating that PA effectively increased the expression of phosphorylation of ACC and ATGL in EC tissue (Figure 8I and Appendix A). Overall, these results indicate that although PA does not increase body weight, it increases lipogenesis in tumor tissue after 4 weeks of treatment. The mechanism by which PA inhibits tumor growth in vivo involves increasing cellular stress and apoptosis, reducing angiogenesis, and inhibiting AKT/mTOR and MAPK pathway activity.

## 4. Discussion

FAs have been implicated in the progression and prognosis of EC [39]. However, conflicting reports exist about the effect of PA on cell proliferation, adhesion, migration, invasion, tumor growth, and fatty acid metabolism in multiple cancers [18]. Recent studies have demonstrated that PA has anti-tumorigenic potential by targeting multiple cell signaling and metabolic pathways in vitro and in vivo [18,25,34,40]. While the relationship between metabolic abnormalities and carcinogenesis and progression of EC is well established, the critical importance of FAs, including PA, for cell proliferation and tumor growth in EC has been less well studied. Using EC cell lines and a transgenic mouse model of EC, we found that physiological concentrations of PA significantly reduced cell viability, caused cellular stress and apoptosis, decreased cell invasion and migration, and inhibited tumor growth through inhibition of the AKT/mTOR and MAKP signaling pathways in vitro and in vivo. EC cells increased intracellular fatty acid synthesis and LD formation in response to different extracellular concentrations of PA. LDs have the potential to protect EC cells from PA-induced cytostatic effects. Cellular stress was a trigger to control cell proliferation, apoptosis, and invasion in PA-treated EC cells. Additionally, seventeen primary cell cultures from human EC tissues were used to examine the effect of different concentrations of PA on cell proliferation, which showed that these primary cultures responded differently to 250 and 500 µM PA treatments in the MTT assay. Although we did not find any correlation between pathological grade, clinical stage, pathological diagnosis, and PA sensitivity in the primary cultures, we speculate that these differences in response to PA may be related to different molecular classifications and metabolic status of ECs. Overall, these results highlight the role of PA in cell viability and tumor growth in EC cells and the *Lkb1^fl/fl^p53^fl/fl^* mouse model of EC, respectively.

Invasion and migration of EC cells are initiated and mediated by multiple signaling pathways that control the processes of EMT, including integrins and matrix-degrading enzymes, which promote cell detachment from the primary tumor and the consequent ability of cells to procure a motile phenotype through cell–matrix interactions [41,42]. The reprogramming of lipid metabolism potently changes the fatty acid composition of the membrane, enhances cell membrane fluidity, and promotes EC progression through EMT signaling pathways [43,44]. Increasing evidence suggests that PA has anti-invasive and anti-metastatic impacts on certain types of cancer cells [45]. In gastric cancer, low concentrations of PA (50 and 100 μM) induced cell migration and invasion through CD36-dependent activation of the AKT pathway, whereas high concentrations of PA reduced cell migration and invasion possibly because of lipotoxicity [46]. In contrast, 10 or 20 μM PA exhibited an inhibitory effect on the migration and invasion in PC3 and DU145 prostate cancer cells, at least partly by regulating PKCζ, integrin β1, and EMT processes, and abrogated M2 macrophage-induced EMT and migration of colorectal cancer cells. Furthermore, 100 µM PA significantly promoted cell proliferative activity, migratory capacity, and EMT process in PC3 cells [34,40,47,48]. Given the recently identified association of the dual effects of PA on invasive capacity in various cancer cells, we initially focused on the influence of PA on cell motility by employing adhesion, modified Boyden chamber, and wounding healing assays in EC cells. Low concentrations of PA did not increase or decrease cell adhesion or invasion, while PA concentrations over 100 µM did reduce EC cell adhesion and invasion through the inactivation of MMP9 and a reduction in the EMT process. IHC results confirmed that PA treatment decreased the expression of VEGF-R2 in EC tissues. The inhibition of ROS by NAC resulted in partial antagonism of PA-induced cell migration in both cell lines. Our results, together with previous observations in other laboratories, suggest that the effects of PA on migration and invasion may depend in part on tumor type and concentration, and PA-induced cellular stress is a trigger for cell adhesion and invasion in EC cells. The underlying regulatory mechanisms need to be explored in further experiments.

High concentrations of free FAs, particularly saturated FAs, including PA, result in an imbalance between ER protein load and ER folding, ultimately triggering ER stress, apoptosis, and death in cancer cells [49,50]. Palmitoylation has been linked to ER stress by regulating trafficking, localization, stability, and aggregation in ER [51]. Excessive intake of PA can directly or indirectly affect the accumulation of intracellular ROS and induce cellular stress, and its effects may be related to cell type [49,52]. Exogenous PA has the ability to alter the lipid composition in the ER of MDA-MB-231 breast cancer cells, and the lipid composition in the ER membranes appears to be involved in the inhibition of PA-induced cell proliferation and apoptosis [53]. Increasing evidence shows that the generation of cellular stress in response to PA treatment has been implicated as an important contributor to PA-induced apoptosis in cancer cells. Inhibition of cellular stress significantly reduces PA-induced apoptosis in cardiomyocytes, Leydig cells, pre-frontal cells, and colon carcinoma cells [54,55,56,57]. In addition to cellular stress, PA-mediated apoptosis appears to involve a number of relevant factors and signaling pathways including p53, p62, p21, Sesn2, IDH1, CD36, S6K1, and the PI3K/Akt pathways [34,58,59,60,61]. In our studies, we found that both the Ishikawa and ECC-1 cell lines exhibited different sensitivity to the induction of ROS production and reduction in mitochondrial membrane potential when treated with PA above 100 µM. This difference in the response of EC cell lines to PA-induced cellular stress may be further delineated by characterizing the distinct lipid profile and metabolism of each cell subset, as varying EC cell lines exhibit different glucose and lipid metabolic profiles [62]. Our previous work showed that cell proliferation of Ishikawa and ECC-1 cells predominantly depended on glycolysis through multiple complex signaling pathways compared with other EC cell lines [63]. In addition, we found that PA is responsible for inducing the intrinsic and extrinsic pathways of apoptosis in both cell lines regardless of the differences in the cell stress responses of the two cells to PA. The inhibition of cellular stress by NAC partially attenuated apoptosis and increased cell viability in both cell lines. Overall, we believe that multiple factors ranging from genetic background and metabolic profile to lipid composition in ER membranes may be factors affecting PA-induced cellular stress in EC cells. Cellular stress is a trigger that controls PA-induced apoptosis and cell proliferation inhibition in EC cells.

LDs are emerging as novel regulators of cell growth, metabolism, invasion, inflammation, immunity, neoplastic transformation, and drug resistance. The accumulation of LDs is generally considered to be beneficial to the survival of cancer cells because of the fact that it can sequester inflammatory-inducing lipids such as ceramides and DAGs, safeguard polyunsaturated FAs from lipid peroxidation that causes DNA damage, and protect cancer cells under stress and oxidative stress [64,65]. Given that excess intracellular LDs have been shown to be substrates for autophagy, autophagy, and, especially, lipophagy, it has recently been implicated to be responsible for regulating LD formation by mobilizing triglycerides and cholesterol with acid hydrolases in LDs to maintain lipid homeostasis [66,67]. However, excessive FA overload loading or impeded lipid lipolysis leads to excess accumulation of LDs, eventually resulting in lipotoxicity in normal and cancer cells [65,68,69]. In gastric cancer cells, treatment with FAs or adipocytes induced intracellular LD formation through transcriptional upregulation of DGAT2 in a C/EBPα-dependent manner [70]. Similarly, treatment of HepG2 cells with 200 μM PA for 24 h resulted in the formation of LDs and stimulated mitochondrial oxidative metabolism [71]. Interestingly, Raman microscopic observations found that the average number and size of LDs in PA-treated cells were lower than those LDs in cells treated with other FAs, suggesting that FA types may have varying influences on the formation of LDs [72]. Inhibition of TAG synthesis by targeting DGAT1 and DGAT2 with specific small inhibitors effectively depleted the formation of LDs and had differential effects on cell growth depending on the type of FA in cancer cells [38,73,74]. In the current study, increasing PA concentrations increased the formation of LDs in a time- and dose-dependent manner in the EC cell lines. Blocking the accumulation of LDs by DGTA1/2 inhibitors effectively enhanced the ability of PA to inhibit cell proliferation in both EC cell lines. Since DGAT1 and DGAT2 are enzymes that catalyze the last step in TAG synthesis and control the biogenesis of LDs, inhibition of their functions results in an increased concentration of free FAs and other lipid species in the cytoplasm, ultimately leading to higher “lipotoxicity” for cancer cells [38,75]. Recent studies confirmed that DGAT1-deficient cells were sensitized to lipid stress, resulting in increased caspase 3 and 7 activation, and knock-down of DGAT1 by siRNA decreased LD formation, resulting in autophagy and inhibited cell growth in prostate cancer LNCaP cells [75,76]. Thus, pharmacological inhibition of DGAT1/2 may have therapeutic potential in the treatment of cancer, including EC [74].

PA, as an intracellular signaling molecule, participates in the activation and inactivation of various signaling pathways to regulate tumor growth. Inhibition or promotion of tumor cell growth by PA often depends on multiple pathways with different functions, without any specific signaling pathway being involved [18]. For example, PA inhibited cell proliferation and M2-TAM-induced migratory and invasive properties by inhibiting the IL-10-STAT3-NF-κB signaling axis in colorectal cancer cells [40]. On the other hand, treatment of neuroblastoma cells with 200 µM PA did not affect cell viability but blocked insulin-induced metabolic activation, inhibited the activation of the insulin/PI3K/Akt pathway, and activated mTOR kinase [77]. Interestingly, sarcoma cells exposed to 400 µM PA promoted cell proliferation and activated the AKT/mTOR/S6 pathway through the phosphorylation of PTEN at T366 [78]. Our results showed that 250 µM PA significantly decreased the expression of phosphorylated AKT, S6, 4-EBP1, p38, and p42/44 in the Ishikawa and ECC-1 cells and inhibited the expression of phosphorylation of S6 and p42/44 in PA-treated tumor tissues. These results support the fact that PA inhibits tumor growth involving AKT/mTOR and MAPK pathways in EC.

A few studies have confirmed that PA exhibits opposite effects on cancer cell proliferation and invasion in different types of cancer, and even within the same type of cancer, PA has different effects on tumor growth in preclinical models of cancer [18,25,79]. Given that PA either serves as an energy source for cancer cell growth or as a signaling molecule that regulates cancer cell growth, it is difficult to explain these phenomena in terms of tumor type. Metabolic reprogramming, cancer phenotypes, the homeostatic balance of PA, and the PA composition of the cell membrane may be responsible for these differences. In addition, although PA plasma concentrations in healthy subjects vary widely, ranging from 0.1 to 4.1 mmol/L, the doses we used in this study were 100 and 250 μM, which are within the physiological concentration range of PA [18,80,81]. The treatment of EC cells with 100 and 250 μM PA significantly increased cellular LD formation and TG concentration, induced cellular stress, and caused apoptosis. The *LKB1^fl/fl^ p53^fl/fl^* mice treated with 10 mg/kg PA for 4 weeks did not increase serum TG concentration, body weight, or liver weight, but balloon-like hepatocytes were shown in the treatment group. These results indicate that these doses of exogenous PA are effective in producing changes in biological function in vitro and in vivo. A similar study also suggested that 100 and 200 μM PA significantly induced apoptosis, DNA damage, and cell cycle arrest in EC cell lines RL95-2 and HEC-1A [80]. Furthermore, a metabolomic analysis of blood samples found significantly reduced PA concentrations in patients with stage I and II EC compared with healthy women [82]. Overall, based on current results and the few other studies on PA in EC, we believe that physiological concentrations of exogenous PA can act as a tumor suppressor through various biologic pathways, participating in inducing cellular responses and apoptosis to inhibit cell proliferation and apoptosis in EC. The effects of different doses of PA and different treatment durations as well as the combined use of PA with other FAs on lipid pool balance and cell membrane composition deserve further study in our EC cell lines and *LKB1^fl/fl^ p53^fl/fl^* mouse model of EC.

In addition to the antitumor activity of PA, recent studies have demonstrated that the combination of PA with chemotherapeutic drugs or biomolecules can significantly increase the sensitivity of cancer cells to these agents [83]. PA increased the sensitivity to methylseleninic acid-induced apoptosis and inhibition of tumor growth by enhancing CHOP in a HepG2 xenograft model [33]. Nanoparticle-encapsulated PA combined with doxorubicin reduces cell proliferation, tumor growth, and metastatic capacity in breast cancer cells and mouse models compared with monotherapy [84]. Moreover, physiological concentrations of PA increase the sensitivity of cisplatin and doxorubicin to growth inhibition of human EC cells [80]. Although the mechanism underlying the increased sensitivity of PA to these drugs is currently unclear, it will be worthwhile to evaluate combinations of PA with cisplatin and paclitaxel in EC cells and *LKB1^fl/fl^ p53^fl/fl^* in our upcoming projects.

## 5. Conclusions

There are limited data on the effect of FAs on cell proliferation and tumor growth in EC. We report that PA exhibited anti-proliferative and anti-tumorigenic activities in EC cells and the *Lkb1^fl/fl^p53^fl/fl^* mouse model of EC through the induction of cellular stress and apoptosis and the inhibition of AKT/mTOR and MAPK pathways. Cellular stress is a key component mediating the effects of PA on EC cell growth, apoptotic invasion, and tumor growth. PA in combination with targeting the formation of LDs may provide new insights into the treatment of EC. Future studies are necessary to explore the anti-tumor mechanisms of PA further, as well as its effects in combination with other chemotherapeutic drugs such as carboplatin, laying the foundation for future clinical studies of this novel dietary supplement in obesity-driven EC. 

## Figures and Tables

**Figure 1 biomolecules-14-00601-f001:**
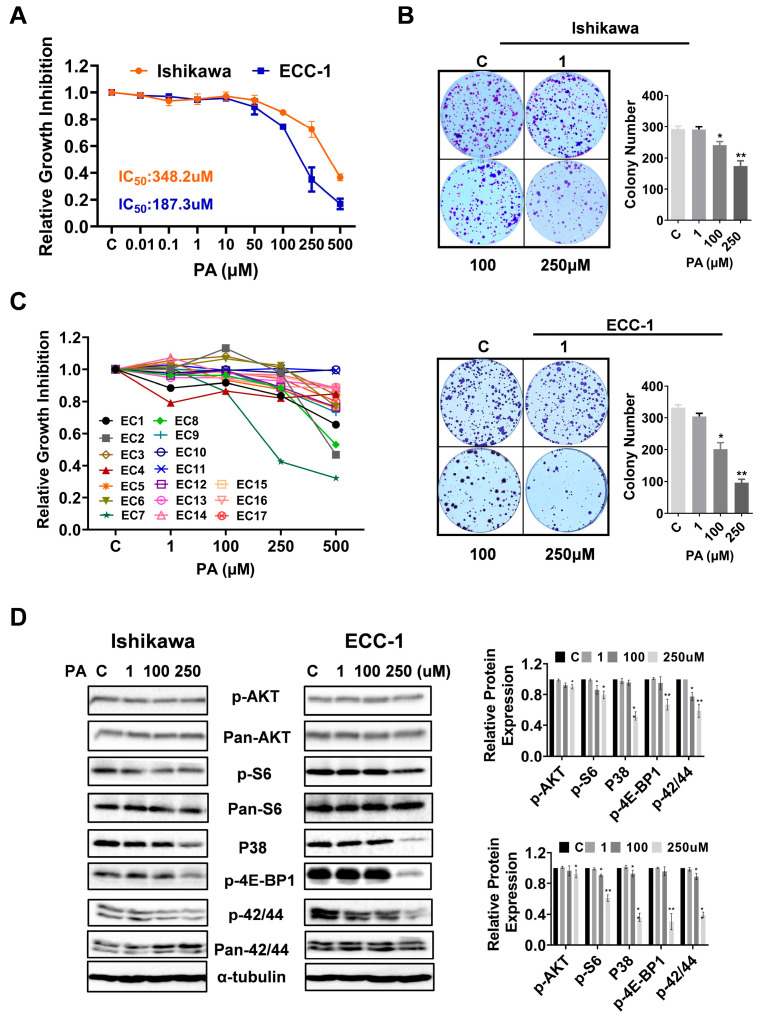
PA inhibited cell proliferation in EC cells and primary cultures of human ECs. Cell proliferation and IC50 values of Ishikawa and ECC-1 cells treated with PA at the indicated concentrations for 72 h (**A**). Colony formation of Ishikawa and ECC-1 cells treated with PA (**B**). Cell proliferation in 3 primary cultures of human ECs treated with PA at the indicated concentrations for 72 h (**C**). Results for the remaining 14 samples are shown in Appendix A. Expression of phosphor-AKT, S6, 4E-BP-1, p38, and p42/44 proteins in Ishikawa and ECC-1 cells, and quantification of the expression levels (**D**). * *p* < 0.05, ** *p* < 0.01. Original images of (**D**) can be found in Appendix A.

**Figure 2 biomolecules-14-00601-f002:**
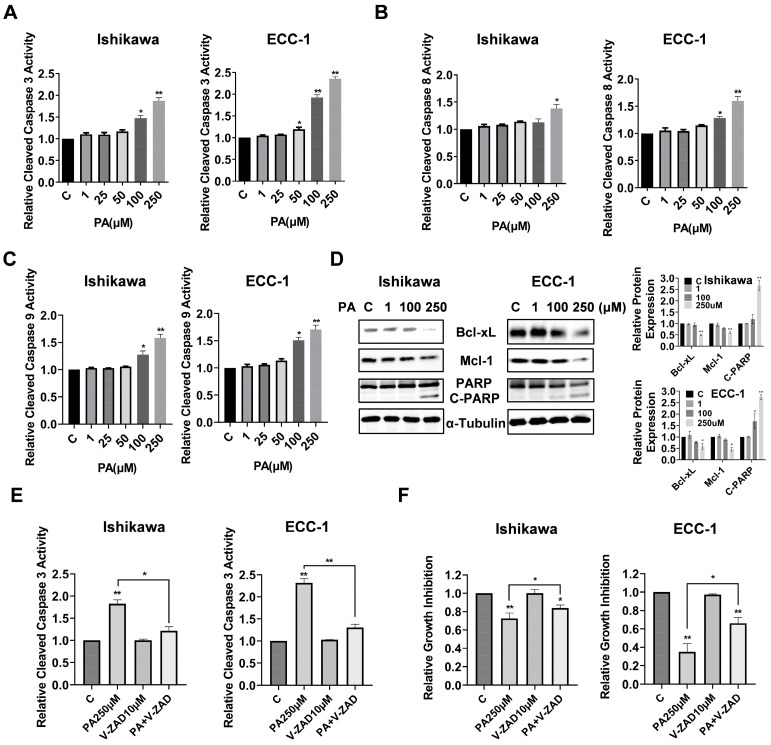
PA induced apoptosis in Ishikawa and ECC-1 cells. Cleaved caspases 3, 8, and 9 levels in Ishikawa and ECC-1 cells treated with PA for 14 h (**A**–**C**). Expression of Bcl-xL, Mcl-1, and cleaved PARP in Ishikawa and ECC-1 cells treated with PA for 14 h, and quantification of the expression levels (**D**). Changes in PA-induced cleaved caspase 3 activity after pre-treatment with Z-VAD-FMK in both cells (**E**). Changes in cell viability induced by PA after pre-treatment with Z-VAD-FMK in both cells (**F**). * *p* < 0.05, ** *p* < 0.01. Original images of (**D**) can be found in Appendix A.

**Figure 3 biomolecules-14-00601-f003:**
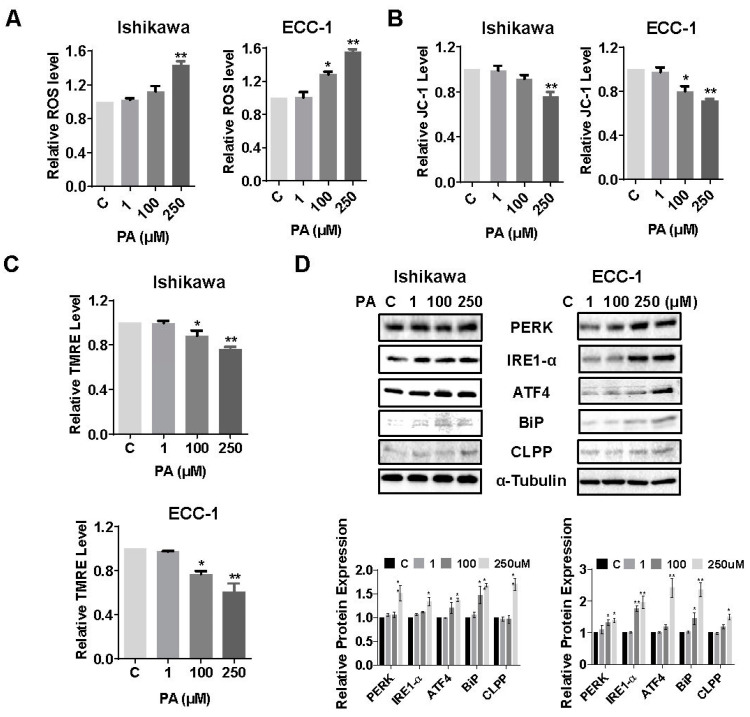
PA induced cellular stress in Ishikawa and ECC-1 cells. Effect of PA on ROS levels in both cell lines (**A**). Changes in JC-1 and TMRE levels in both cell lines after PA treatment (**B**,**C**). Protein expression of cellular stress-related proteins, including PERK, IRE-1α, ATF4, BiP, and CLPP, in Ishikawa and ECC-1 cells after PA treatment (**D**). * *p* < 0.05, ** *p* < 0.01. Original images of (**D**) can be found in Appendix A.

**Figure 4 biomolecules-14-00601-f004:**
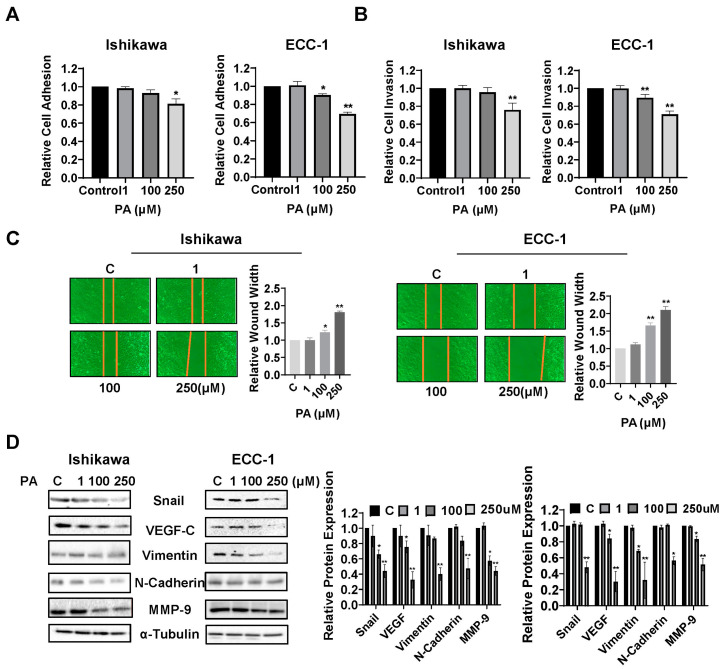
PA inhibited adhesion and invasion in Ishikawa and ECC-1 cells. Effect of PA on adhesive ability in the Ishikawa and ECC-1 cells (**A**). A transwell assay was used to investigate invasive ability in both cell lines after PA treatment (**B**). Migratory abilities of Ishikawa and ECC-1 cells after 48 h of PA exposure (**C**). Protein expression of EMT-related proteins, including Snail, VEGF, Vimentin, N-Cadherin, and MMP-9, in Ishikawa and ECC-1 cells treated with PA (**D**). * *p* < 0.05, ** *p* < 0.01. Original images of (**D**) can be found in Appendix A.

**Figure 5 biomolecules-14-00601-f005:**
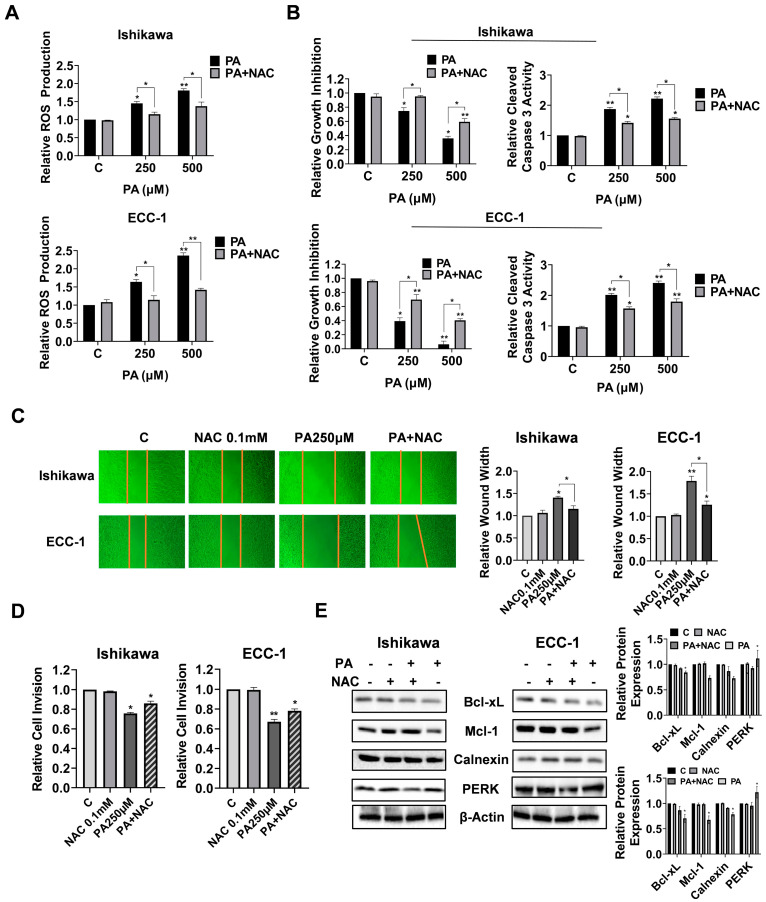
Cell stress modulates the inhibition of cell migration and invasion by PA in Ishikawa and ECC-1 cells. Changes in PA-induced ROS production after pre-treatment with 0.1 mM NAC in Ishikawa and ECC-1 cells (**A**). Effects of pre-treatment with 0.1 mM NAC on PA-induced cell viability and apoptosis in both cell lines (**B**). Effects of pre-treatment with 0.1 mM NAC on PA-induced cell migration and invasion in both cell lines (**C**,**D**). Expression of Bcl-xL, Mcl-1, Calnexin, and PERK in both cells after pre-treatment with NAC and quantification of the expression levels (**E**). * *p* < 0.05, ** *p* < 0.01. Original images of (**E**) can be found in Appendix A.

**Figure 6 biomolecules-14-00601-f006:**
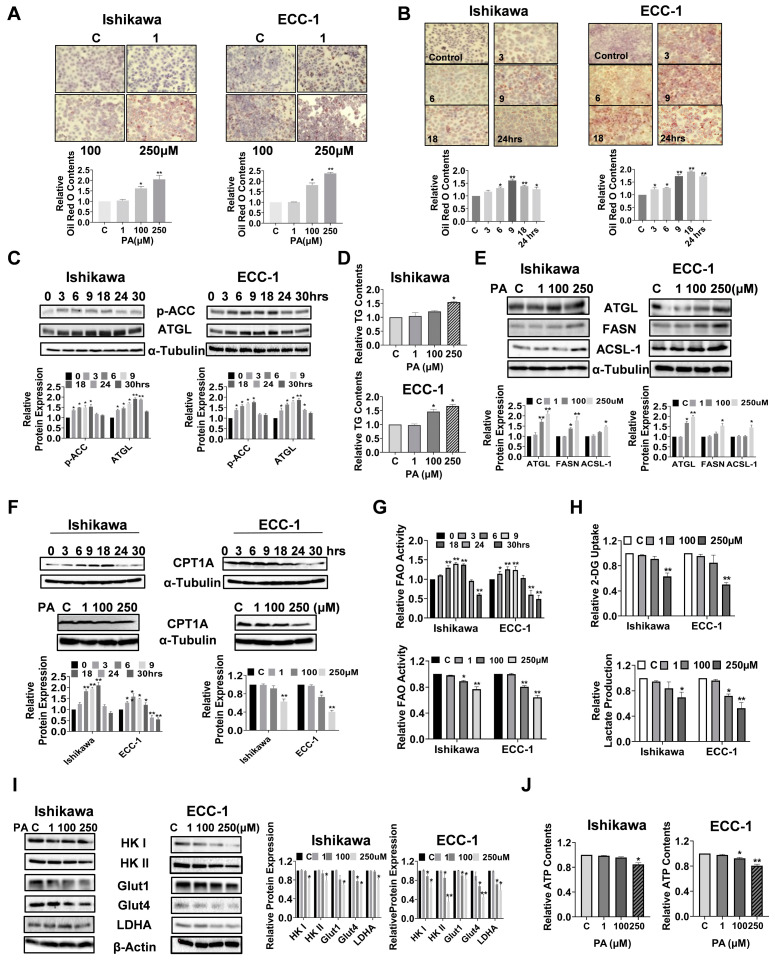
PA increased lipid droplet (LD) formation and lipogenesis in Ishikawa and ECC-1 cells. Formation of LDs after treatment with PA in a dose-dependent and time-dependent manner in both cell lines (**A**,**B**). Protein expression of p-ACC and ATGL after PA treatment in Ishikawa and ECC-1 cells (**C**). Changes in intracellular TG levels in both cell lines treated with 250 μM PA (**D**). Protein expression of ATGL, ACSL-1, and FASN in both cell lines following 24 h of PA or placebo treatment, and quantification of the expression levels (**E**). Protein expression of CPT1 after treatment with PA in a time-course and dose-dependent manner in both cells (**F**). Changes in FAO activity after treatment of both cells with PA in a time-course and dose-dependent manner (**G**). Effects of PA on glucose uptake and lactate production in both cell lines (**H**). Effects of PA on the protein expression of Glut1, Glut4, LDHA, Hexokinase I, and Hexokinase II in both cells (**I**). Changes in cellular ATP production in both cells after PA treatment (**J**). * *p* < 0.05, ** *p* < 0.01. Original images of (**C**,**E**,**F**,**I**) can be found in Appendix A.

**Figure 7 biomolecules-14-00601-f007:**
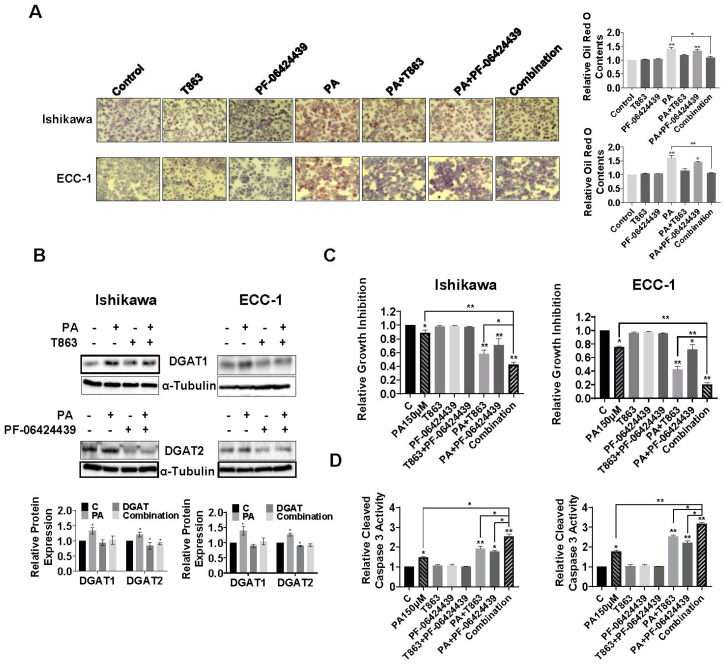
Inhibition of LD formation increased the sensitivity to PA in Ishikawa and ECC-1 cells. Effects of DGAT1 inhibitor (T863) and DGAT2 inhibitor (PF-06424439) on LD formation induced by PA in both cells (**A**). Protein expression of DGAT1 and DGFAT2 in both cells after treatment with 150 µM PA, 5 µM T863/ PF-06424439, or the combination for 24 h (**B**). Effects of T863 and PF-06424439 on PA sensitivity in inhibition of cell proliferation (**C**) and induction of cleaved caspase 3 activity in both cells (**D**). * *p* < 0.05, ** *p* < 0.01. Original images of (**B**) can be found in Appendix A.

**Figure 8 biomolecules-14-00601-f008:**
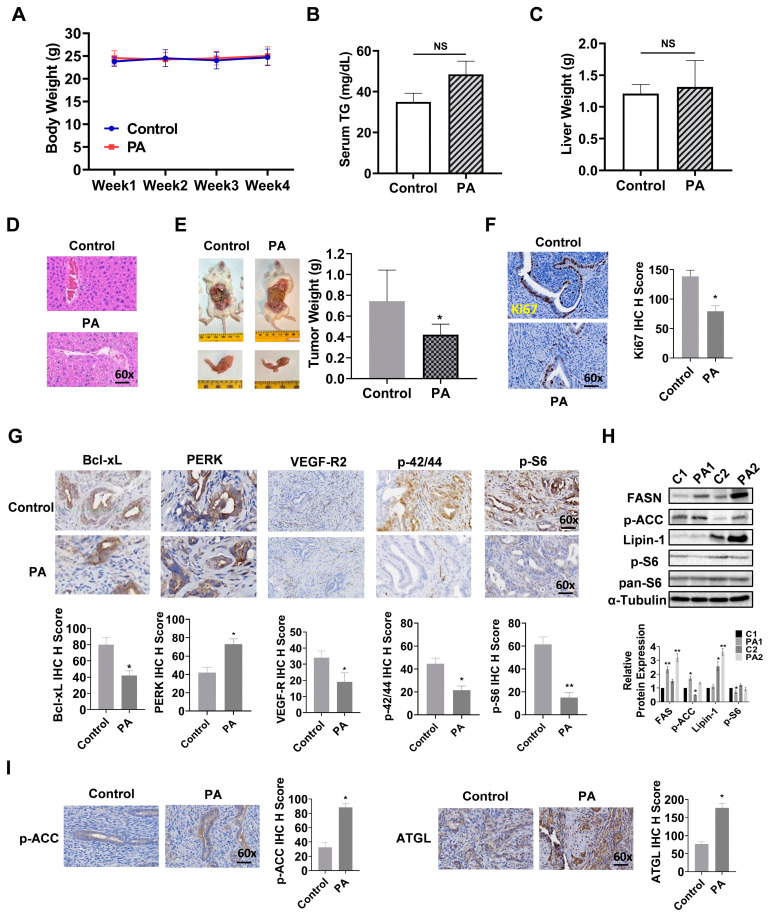
PA inhibited tumor growth in the *Lkb1^fl/fl^p53^fl/fl^* transgenic mouse model of EC. Body weight of *Lkb1^fl/fl^p53^fl/fl^* mice during PA treatment (**A**). Changes in serum triglycerides (TGs) and liver weights after 4 weeks of PA treatment (**B**,**C**). H&E staining of liver sections from PA-treated and control mice (**D**). Tumor weights in *Lkb1^fl/fl^p53^fl/fl^* mice after 4 weeks of PA treatment (**E**). Protein expression of Ki-67 in EC tissues from *Lkb1^fl/fl^p53^fl/fl^* mice following PA or placebo treatment (**F**). Effects of PA on the expression of Bcl-xL, PERK, VEGF-R2, p-42/44, and p-S6 in EC tissues (**G**). The protein expression of FASN, p-ACC, Lipin-1, and p-S6 in EC tissues. C1 and C2 are untreated EC tissues. PA1 and PA2 are PA-treated EC tissues (**H**). Effects of PA on the expression of p-ACC and ATGL in EC tissues from *Lkb1^fl/fl^p53^fl/fl^* mice (**I**). * *p* < 0.05, ** *p* < 0.01, NS: Non-significant. Original images of (**H**) can be found in Appendix A.

## Data Availability

The raw data supporting the conclusions of this article will be made available without reservation by the authors.

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
