# Peer review of "Palmitic Acid Exerts Anti-Tumorigenic Activities by Modulating Cellular Stress and Lipid Droplet Formation in Endometrial Cancer"

_biomolecules, 2024, doi:10.3390/biom14050601_

Round 1

Reviewer 1 Report

Comments and Suggestions for Authors

I did not see much improvements in the revised version that still has several flaws not convincing

Author Response

Authors’ point-to-point responses to report 1:

3/27/2024

Biomolecules

Dear Reviewer,

We have submitted via the internet our revised manuscript entitled “Palmitic acid inhibits cell proliferation and tumor growth of endometrial cancer in vitro and in vivo”. We would like to thank you and the reviewers for your insightful comments, which have greatly helped us to improve the quality of our manuscript. It is our belief that the manuscript is substantially improved after making the suggested edits.

In the revised manuscript, we have responded to your questions, including removing a figure of tumor volume and explaining PA dose, etc. We have highlighted these changes in red (we also keep previous changes in red). We hope that these revisions are sufficient to make our manuscript suitable for publication in Biomolecules.

Thank you for your consideration and acceptance of our manuscript.

Sincerely,

The Authors

Reponses:

Comments and Suggestions for Authors:

The manuscript reports the activity of PA in endometrial cells in vitro and in vivo. While there is a deep analysis of the potential effects of PA on endometrial cancer cells, the true efficacy and the interpretation of the results are, in the opinion of this Reviewer , are not convincing.

  1. The effects reported in the two EC cell lines are biologically relevant only considering 250 micromola, that is true is a concentration achievable but nonethelss is a quite high concentration. In addition, in the human derived samples, a part from one, all the others have little or no effects at 250 micromolar, that again questions about the true efficacy of

Thanks for the comments. Plasma concentrations of PA vary widely from 0.1 to 4.1 mmol/L in health subjects, depending on the method of measurement(1, 2). Many recent research papers report that 100-250 uM should be a physiological concentration of PA. PA at 100-250 uM has significant effects on cell proliferation in multiple cancer cell lines(3-7). Of our 17 EC primary cultures, 3 did not respond to PA treatment and the other 3 responded well to PA treatment, achieving 50% inhibition of cell proliferation at 250 or 500 uM PA. The remaining samples showed at least 10% inhibition of cell proliferation at 250 uM PA (supplemental figure 1). The primary cultures we used in this study were derived from collagenase digestion and may contain 10% to 20% stromal cells, which may affect sensitivity of PA. In addition, based on many years of experience using primary cultures in our lab, even highly pure primary cultures are less sensitive to chemotherapeutic drugs compared with EC cancer cell lines. Our animal study showed that treatment of LKB1 p53 mice with PA (10 mg/kg) for 4 weeks effectively reduced tumor weight and tumor volume. Therefore, we believe that 250 or 500 uM PA should be considered sensitive to growth inhibition of EC cell lines and EC primary cultures.

  1. IN vivo it is not clear what difference exisrts between volume and It has not been described how these two parameters are calculated and is difficult to understand why the effect on tumor weight is limited compared to that observed when the volume is considered.

Thanks for the comments and suggestions. In this study, after 4 weeks of treatment, PA inhibited tumor volume and reduced tumor weight in the transgenic mouse model of EC. Because tumor volume is a good proxy for tumor weight, we present tumor volume data in addition to tumor weight in Figure 8. Based on our experience with animal work, we believe that tumor weight is a better predictor of tumor response to treatment than tumor volume because the shape of endometrial tumors is irregular when we harvest them, making it sometimes difficult to calculate volume, especially measuring tumor width. This is why in this study, although PA significantly reduced tumor volume and tumor weight, there was no good agreement between tumor weight and tumor volume. After discussion with our team, we decided to remove the tumor volume data in Figure 8 to avoid confusion for readers.

  1. Also for the remaining experiments a small effect is observed at 250 micromolar and sometimes the statistical significance does not convince in terms of biological significance of the differences

Thanks for the comments. As we mentioned in question 1, 250 uM PA significantly produced a series of biological roles in inhibiting cell proliferation, causing cellular stress and apoptosis, and reducing invasive ability in EC cells. Sometimes, a small effect does not mean it can't elicit a significant biological response in an experimental treatment. For example, in some of our experiments, a 10-20% increase in ROS was sufficient to induce apoptosis and inhibit cell growth in our experimental system. In addition, we have previously evaluated several antitumor drugs, such as PARP inhibitors and dopamine receptor D2 (DRD2) antagonists, which did not show much biological response (10-30%) even at high doses in our experimental system.

  1. The possible use of PA should be carefully considered given that in some tumors as highlighted by the authors, it has opposite effects.

We completely agree with the reviewer's comments. Some kinds of fatty acids including PA exhibit opposite effects on tumor cell proliferation and tumor growth in different cancer types. Even within the same type of cancer, PA affects cell proliferation differently. We think it is difficult to explain these phenomena in terms of tumor type. Different metabolic reprogramming backgrounds and lipid composition of cell membrane in cancer cells may be responsible for these differences. This possibility deserves further investigation. Although our study demonstrates that PA has antitumor activity in EC, much, much more work remains to be done before designing clinical trials in EC patients.

  1. The authors alluded at a possible combination with other agents. This important point should be checked to give more biological relevance to the

Among the anti-tumor activities of PA, PA can enhance the synergistic effect on chemotherapy drugs in certain types of cancer cells. In this study, we found that inhibition of DGAT1/2 by small molecule inhibitors significantly increased the sensitivity of PA in inhibiting cell proliferation. These results suggest that PA in combination with other agents may produce more biologic activities in inhibition of cell proliferation and tumor growth. Our next research objective is to investigate how the combination of PA with other fatty acids influences metabolic changes and the composition of cell membranes, in addition to studying the impact of PA with carboplatin or paclitaxel on cell growth and tumor development in EC. We have added some information into the Discussion.

References

  1. Abdelmagid SA, Clarke SE, Nielsen DE, Badawi A, El-Sohemy A, Mutch DM, et al. Comprehensive profiling of plasma fatty acid concentrations in young healthy Canadian adults. PLoS One. 2015;10(2):e0116195.
  2. Fatima S, Hu X, Gong RH, Huang C, Chen M, Wong HLX, et al. Palmitic acid is an intracellular signaling molecule involved in disease development. Cell Mol Life Sci. 2019;76(13):2547-57.
  3. Lin L, Ding Y, Wang Y, Wang Z, Yin X, Yan G, et al. Functional lipidomics: Palmitic acid impairs hepatocellular carcinoma development by modulating membrane fluidity and glucose metabolism. Hepatology. 2017;66(2):432-48.
  4. Yu G, Luo H, Zhang N, Wang Y, Li Y, Huang H, et al. Loss of p53 Sensitizes Cells to Palmitic Acid-Induced Apoptosis by Reactive Oxygen Species Accumulation. Int J Mol Sci. 2019;20(24).
  5. Wang P, Lu YC, Wang J, Wang L, Yu H, Li YF, et al. Type 2 Diabetes Promotes Cell Centrosome Amplification via AKT-ROS-Dependent Signalling of ROCK1 and 14-3-3σ. Cell Physiol Biochem. 2018;47(1):356-67.
  6. Wu ZS, Huang SM, Wang YC. Palmitate Enhances the Efficacy of Cisplatin and Doxorubicin against Human Endometrial Carcinoma Cells. Int J Mol Sci. 2021;23(1).
  7. Eynaudi A, Díaz-Castro F, Bórquez JC, Bravo-Sagua R, Parra V, Troncoso R. Differential Effects of Oleic and Palmitic Acids on Lipid Droplet-Mitochondria Interaction in the Hepatic Cell Line HepG2. Front Nutr. 2021;8:775382.

Reviewer 2 Report

Comments and Suggestions for Authors

The revised paper looks good. The authors have made the required revision. The manuscript contains many valuable results and findings obtained using a range of complementary methods, and deserves to be published. However, I would recommend minor revision before publication.

Figures 1F, 1H, 3E, 4E, 2G, 7A and 8B contains various microphotgraphs. A magnification or scale bar should be added to the captions or to corresponding Figures. 

Author Response

Review report 2:

The revised paper looks good. The authors have made the required revision. The manuscript contains many valuable results and findings obtained using a range of complementary methods, and deserves to be published. However, I would recommend minor revision before publication.

Figures 1F, 1H, 3E, 4E, 2G, 7A and 8B contains various microphotgraphs. A magnification or scale bar should be added to the captions or to corresponding Figures.

Authors’ point-to-point responses to report 2:

Answer: Thanks for the comments. We have added the information to the figures.

Reviewer 3 Report

Comments and Suggestions for Authors

I have previously reviewed the initial version of the manuscript of Zhao Ziyi and colleagues, and the revised version has been significantly improved. I’d like to confirm high novelty and practical importance of the results. However, several of my previous comments require additional attention:

-          The authors used varying exposition time in individual assays/analyses and, sometimes, different PA concentrations (150 um in experiments with DGAT inhibition). Since both the factors are very important to registered effects, I recommend adding an explanation (where appropriate) for the choice of experimental conditions other than those typically used in this study. Also, the reason of using the dose 10 mg/kg PA in animal experiment must be commented.

-          The style of the Discussion is more suitable for a Review article. It is overloaded with plain detailed description of the previously published works, but does not provide the explanation of the cause-and-effect relationship between the results obtained in the study. I pointed at this drawback previously, but no significant changes were made. I recommend explanation of the PA mode of action starting from molecular mechanisms to cellular and physiological phenomena observed.

-          The results of bioinformatics analysis are presented in SI only, and described in Discussion without mentioning in Results. It is impossible to understand the results without and information and even Figure legends. If the authors consider this analysis important to substantiate the main idea of the work, they must include this part of study to the Methods and Results sections and provide a detailed description.

Minors

-          Line 436: Figure 3E presents the data on PERK expression, not BCL-xL.

-          Line 605: The correct phrase is “Abs specific to phosphorylated forms of target proteins”, not “phosphorylated Abs”.

-          Line 669: The statement is now about IHC results on VEGF expression in EC tissues. However, the study was performed for VEGF-R, not VEGF.

Comments on the Quality of English Language

English is good.

Author Response

Review report 3:

I have previously reviewed the initial version of the manuscript of Zhao Ziyi and colleagues, and the revised version has been significantly improved. I’d like to confirm high novelty and practical importance of the results. However, several of my previous comments require additional attention:

-          The authors used varying exposition time in individual assays/analyses and, sometimes, different PA concentrations (150 um in experiments with DGAT inhibition). Since both the factors are very important to registered effects, I recommend adding an explanation (where appropriate) for the choice of experimental conditions other than those typically used in this study. Also, the reason of using the dose 10 mg/kg PA in animal experiment must be commented.

Answer: We have added dose information (DGAT experiment) in the Results section and mouse dose information in the methods section

-          The style of the Discussion is more suitable for a Review article. It is overloaded with plain detailed description of the previously published works, but does not provide the explanation of the cause-and-effect relationship between the results obtained in the study. I pointed at this drawback previously, but no significant changes were made. I recommend explanation of the PA mode of action starting from molecular mechanisms to cellular and physiological phenomena observed.

Answer: we have worked on the discussion following reviewer’ comments and suggestions, including removing some description of the previously published works and re-organizing paragraphs of cell stress, apoptosis and conclusion.

-          The results of bioinformatics analysis are presented in SI only, and described in Discussion without mentioning in Results. It is impossible to understand the results without and information and even Figure legends. If the authors consider this analysis important to substantiate the main idea of the work, they must include this part of study to the Methods and Results sections and provide a detailed description.

Answer: Thanks for the comments. Academic editor has asked us to remove bioinformatics analysis in the discussion and supplemental files.

Minors

-          Line 436: Figure 3E presents the data on PERK expression, not BCL-xL.

Answer: we have corrected this mistake

-          Line 605: The correct phrase is “Abs specific to phosphorylated forms of target proteins”, not “phosphorylated Abs”.

Answer: we added the information into the manuscript.

-          Line 669: The statement is now about IHC results on VEGF expression in EC tissues. However, the study was performed for VEGF-R, not VEGF.

Answer: we have corrected this mistake

Round 2

Reviewer 1 Report

Comments and Suggestions for Authors

The new version and reply are the same as per previous version

Author Response

Thanks to the reviewers and academic editors for their comments. We have corrected the spelling mistake and changed the title to “Palmitic acid exerts anti-tumorigenic activities by modulating cellular stress and lipid droplet formation in endometrial cancer”.

Once again, on behalf of the author team, I would like to thank the reviewers and academic editors for their help.